

# Ground-based detection of Antarctic clouds: analysis of cycles and comparison with IASI products

Federico Donat[1,3], Tiziano Maestri[1], Elisa Fabbri[1], Michele Martinazzo[1], Giovanni Bianchini[2], Massimo Del Guasta[2], Gianluca Di Natale[2], Luca Palchetti[2], Guido Masiello[3], Carmine Serio[3], and Giuliano Liuzzi[3]

[1]Department of Physics and Astronomy, Alma Mater Studiorum - University of Bologna, Bologna, Italy
[2]Department of Engineering, University of Basilicata, Potenza, Italy
[3]Istituto Nazionale di Ottica, Consiglio Nazionale delle Ricerche, Italy

**Correspondence:** Tiziano Maestri (tiziano.maestri@unibo.it)

**Abstract.** Over Antarctica, the identification of cloud layers from infrared satellite observations is extremely challenging due to the similarities in temperatures and radiative properties of the clouds and the underlying iced surfaces. Ground-based observations, collected by the Radiation Explorer in the Far InfraRed - Prototype for Applications and Development (REFIR-PAD) spectroradiometer operating at Concordia Station, Dome C, on the Antarctic Plateau are used to obtain scene classifications

with the Cloud Identification and Classification (CIC) algorithm. The resulting cloud occurrence time series span the timeframe 2014-2020 showing cycles of 12 months (with maxima in December) and 6 months (with maxima in January and July), providing evidence of the semiannual oscillation of the Southern Hemisphere also in localized cloud occurrence. Similar harmonics are observed in the collocated surface temperature and pressure. Analysis of the cloud radiative effect shows that the far infrared downwelling radiance during the peaks of the semiannual oscillation is about twice as high as during its minima. Ground-based

cloud classifications are compared to satellite-derived products of the Infrared Atmospheric Sounding Interferometer (IASI) flying on MetOp A, B, and C. Several IASI L2 cloud products (namely cloud tests, cloudiness summary and cloud phase) collocated with the Concordia Station geolocation are considered. The comparison regards more than 1,200 satellite observations from 2014 to 2020, and is conducted by means of a "one-to-one" correlation analysis and via the analysis of the observed cloud occurrences. The one-to-one analysis (conducted using temporally and spatially collocated measurements from IASI

and REFIR-PAD) shows that, up to December 2019, the IASI products *Artificial Neural Network (ANN)* test and the *Advanced Very-High-Resolution Radiometer (AVHRR) heterogeneity* test are moderately correlated with ground classifications, while the *Numerical Weather Prediction (NWP)* test, *AVHRR cloud fraction* test, and flag *cldnes* are mostly anticorrelated. However, from December 2019, both the *NWP* test and the flag *cldnes* switch to positive correlation values. When the flag *cloud phase* is used as a scene classifier, a limited correlation is found up to December 2019 but significantly higher values are observed in

2020. Finally, it is shown that the IASI cloud phase classification (ice or mixed/liquid) is well correlated with the ground-based phase classification.





# 1 Introduction

Polar regions play a vital role in Earth's energy budget (Liou; Kiehl and Trenberth), as they typically emit more radiant energy
than they absorb from sunlight. In particular, the features of Antarctic clouds (such as cloud amount, thermodynamic phase,
height, optical and microphysical properties) have been shown to influence the global climate system (Lubin et al., 1998). How-
ever, their extensive study is difficult, especially in the interior of Antarctica (Town et al.; Lachlan-Cope; Bromwich et al.).
Satellite-based observation is challenging for short-wave sensors, during the long polar night (King and Turner, 1997), and
for long-wave sensors, because the cloud radiative properties can be similar to those of the icy surface. Furthermore, optically
thin cirrus clouds, whose identification is crucial to accurately retrieving surface properties and atmospheric profiles, are often
present on the Antarctic Plateau (King and Turner, 1997), but their radiative characteristics closely resemble those of clear
skies. While active satellite sensors have been proven valuable for detecting clouds in polar regions, they have limitations,
particularly for satellites with coarse vertical resolution and low sensitivity near the surface, which can hinder the detection of
low-level and geometrically thin clouds (Chan and Comiso; Chan and Comiso).
Ground-based measurements can overcome these difficulties thanks to the high radiative contrast between the cloud and the
cold, non-emitting sky in the infrared atmospheric window (Mahesh et al.; Cox et al.), without the additional variability given
by the surface. These measurements can provide valuable and reliable data and work as benchmarks for satellite-based cloud
classifications and retrievals. Furthermore, the extremely low temperatures and dryness of Antarctica enable the observation of
the far infrared part of the spectrum, which has been proven useful for cloud identification, cloud classification, and the retrieval
of cloud properties (Rathke et al.; Palchetti et al.; Di Natale et al.; Maestri et al.). Unlike satellite observations, ground-based
measurements can also provide continuous, localized data, facilitating the study of short-term cloud variability.

In this study, we utilize a large dataset (more than 230,000 measurements from 2012 to 2020) of ground-based cloud products
derived from high-spectral-resolution acquisitions in the far and mid-infrared by the Radiation Explorer in the Far InfraRed
- Prototype for Applications and Development (REFIR-PAD; Bianchini et al.). This Fourier transform spectroradiometer has
been collecting radiances at Concordia Station, Dome C, on the Antarctic Plateau since 2012. Cloud products are obtained
by processing REFIR-PAD spectra with the Cloud Identification and Classification (CIC) algorithm, which classifies each
spectrum as belonging to a clear sky, ice cloud, or mixed-phase cloud scene. The CIC algorithm has been validated against col-
located lidar measurements (Cossich et al., 2021), and in several simulations tests and field campaigns as summarized in Donat
et al. (2024). A portion of this dataset was previously used in Cossich et al. (2021) to examine cloud occurrence on hourly,
monthly and seasonal timescales, as well as the relative occurrences of ice and mixed-phase clouds. In this paper, we expand on
this work by exploiting the entire dataset for two main purposes. First, by extending the record and applying spectral analysis,
we analyze the frequency of clear and cloudy conditions as a function of time. We find that in addition to the expected annual
cycle (with overall fewer clouds in the cold season and more in the warm season), there is a distinctive semiannual (6 months)
oscillation superposed on the annual cycle. The same harmonic is found in collocated surface temperature and pressure data.
Secondly, we perform a satellite intercomparison between the ground-based cloud detection and cloud information from the
Infrared Atmospheric Sounding Interferometer (IASI) aboard the MetOp satellites. IASI is an infrared sounder that provides





operational Level 2 products including cloud mask and cloud phase. However, over Antarctic sites like Dome C, satellite cloud algorithms face challenges as discussed above. After collocating IASI overpasses with our ground observations, we evaluate how well IASI detects the presence of clouds by means of a scene-by-scene binary (clear/cloud) correlation analysis based

on the Matthews Correlation Coefficient (MCC). This methodology overcomes the issue related to the different fields of view (FOV) of IASI and REFIR-PAD. The IASI FOV is considerably wider than that of REFIR-PAD, and although it includes the footprint of the ground instrument, it may also capture clouds located outside the narrow vertical column observed from the ground. As a result, conflicting scene classifications can occur simply because of the difference in spatial resolution. However, using MCC allows us to verify whether the identification of a cloud (or of a clear sky) from the ground is statistically associated

with a higher (or lower) probability of cloud detection by the satellite, compared to the baseline probability. We also demonstrate that, under simple assumptions, MCC can be used to establish a theoretical upper bound on the satellite's clear/cloud Hit Rate relative to the actual atmospheric state in the satellite's FOV. The polar ground dataset can guide refinements of satellite cloud masks and serve as an independent reference to calibrate or train new cloud detection approaches. Our results emphasize the value of continuous, ground-based remote sensing in Antarctica for both improving climatological understanding of polar

clouds and enhancing the performance of satellite retrievals.

## 2 Data and methodology

### 2.1 Ground-based sensors and products

The cloud products derived from ground based measurements are obtained from data collected by the Radiation Explorer in the Far Infrared–Prototype for Applications and Development (REFIR-PAD). REFIR-PAD is a Fourier transform spectroradiome-

ter (Bianchini et al., 2019) deployed at Dome C as part of the Radiative Properties of Water Vapor and Clouds in Antarctica (PRANA) and Concordia Multi-Process Atmospheric Studies (CoMPASs) projects, under the Italian National Program for Research in Antarctica (PNRA; Palchetti et al.). These projects mark the first extended field campaigns to gather high-spectral-resolution radiances in the FIR, with continuous measurements started in 2012.

REFIR-PAD operates in the broad band range of 100-1500 cm$^{-1}$ in Mach-Zehnder configuration (with two inputs and two

outputs) with a spectral resolution of 0.4 cm$^{-1}$. The calibration is performed by observing two black body sources, the hot blackbody (HBB) stabilised at 80°C and the cold blackbody (CBB) at 10°C. The measurements, each lasting about 60 seconds, are collected in sequences consisting of four atmospheric observations and four calibrations, with two calibrations using the HBB and two using the CBB. Each sequence, lasting approximately 12 minutes, provides four calibrated spectra of the atmosphere for each one of the two output channels. The eight resulting spectra are finally averaged to reduce radiometric noise,

resulting in a single spectrum that represents a 4-minute observation of the atmosphere delivered every 12 minutes. On average REFIR-PAD measures in continuous mode for five hours straight, and then stops for about one hour to make a preliminary processing and transferring of the data to the storage server in Italy. REFIR-PAD is installed in an insulated facility known as the Physics Shelter (PS), in conjunction with a backscattering/depolarization tropospheric lidar (light detection and ranging),





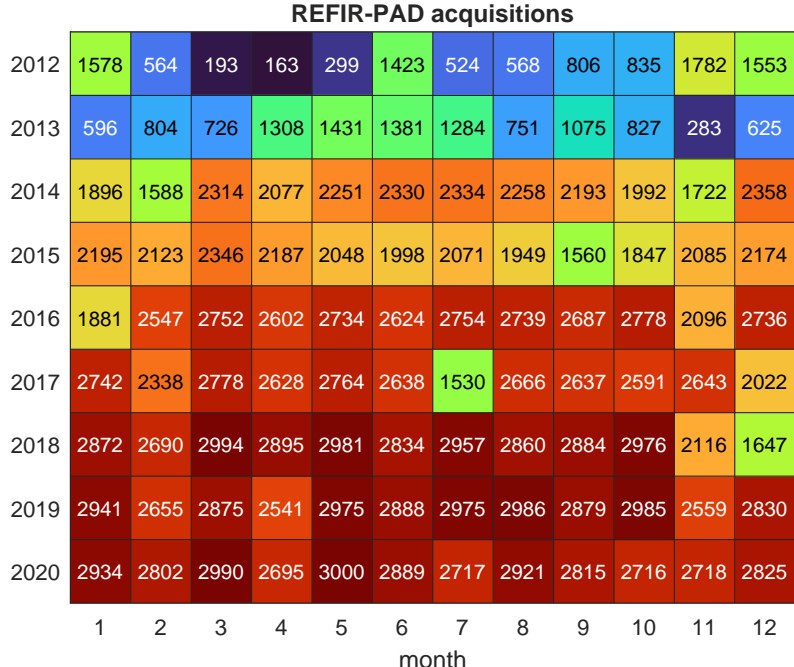

**Figure 1.** Monthly distribution of REFIR-PAD acquisitions from 2012 to 2020.

which detects backscattering and depolarization signals with a temporal frequency of 10 min up to 7 km above the surface

(Palchetti et al., 2015).

The REFIR-PAD available data used for the analysis refer to years from 2012 to 2020. The number of REFIR-PAD observations per month and for each year is reported in Figure 1. The heatmap shows a general increase of the measurements from the first to the last year considered. During year 2012 a positive bias in the number of observations is registered in summer while very few data are available in spring. Also in 2013 the REFIR-PAD observations are few and sparsely distributed along

the course of the year. From 2014 on, at least 1000 observations per month are available and for this reason the time series considered in the analysis ranges from 2014 to 2020 only.

REFIR-PAD spectra are analyzed using an automatic algorithm called Cloud Identification and Classification (CIC). CIC employs a machine learning technique based on principal component analysis (Maestri et al., 2019b) to identify the observed scene from spectrally resolved infrared measurements similarly to what was done in Cossich et al. (2021), where four years of

REFIR-PAD data have been analyzed. In the present work, an improved version of CIC (Donat et al., 2024), with enhanced sensitivity to thin cloud layers and better performance to distinguish ice from mixed-phase clouds, is applied to the series of data. For the warm season (November to March) CIC is set up to identify three classes: clear sky, ice clouds and mixed-



phase clouds. For the cold season (April to October) only the clear sky and ice cloud classes are considered because of the extremely cold temperatures and the negligible occurrence of mixed-phase clouds. For each class and each season a training set of 70 spectra (representative of each year from 2012 to 2020) is created. The true class is provided by the collocated lidar depolarization and backscattering data (Cossich et al., 2021). Only wavenumbers from 380 cm$^{-1}$ to 1000 cm$^{-1}$ are considered, excluding the interval 620-670 cm$^{-1}$ (which is significantly affected by the strong absorption from the $CO_2$ 15 $\mu$m band) as discussed in Maestri et al. (2019b) and Magurno et al. (2020). CIC is trained on these data and its classification performances are tested on separate test sets of 130 spectra each, obtaining an overall clear/cloud Hit Rate of about 98.5%. Finally, CIC is used to classify the remaining spectra in the dataset.

## 2.2 IASI data and collocation

The considered IASI data were downloaded, in 2024, from the EO:EUM:DAT:METOP:IASSND02 collection using the EU-METSAT Data Access Client (EUMDAC) application. IASI data from the satellites Metop A, B, and C were considered. We make use of IASI Level 2 cloud products whose information can be found in the "IASI L2: Product Generation Specification" document (doc. no. EPS.SYS.SPE.990013, V8E, e-signed). We limit our analysis to satellite viewing zenith angles (vza's) smaller than 15° to avoid possible clouds contamination or deviation of the atmospheric state from the zenith of the Dome C station. Also, since IASI fields of view (FOVs) have a 12 km diameter at nadir, we only consider FOVs whose center lies within 6 km from Concordia Station (nominal coordinates -75.10, 123.33). From these IASI spatially collocated data a subset is derived which accounts for a time collocation with a REFIR-PAD measurements. In this case, the constraint is that the satellite measurement occurs within 12 minutes from the beginning of a REFIR-PAD acquisition.

Figure 2 resumes the collocation process and highlights that two types of comparison are possible: 1) the cloud occurrence over Dome C is obtained using IASI L2 products and REFIR-PAD products independently; 2) the IASI L2 products are are also time-collocated with the REFIR-PAD measurements and a correlation analysis is performed.

### 2.2.1 IASI L2 cloud tests

Information regarding the execution and results of cloud tests is stored in the *cldtst* flag. The cloud identification is provided by multiple tests, some of which account for ancillary measurements from the Advanced Very High Resolution Radiometer (AVHRR), also flying on MetOp B and C and operating at different channels simultaneously in the visible and infrared wavelength bands. The considered tests are the following:

– *NWP* test: it is based on a comparison between the measured IASI radiance and simulated clear-sky radiance derived from numerical weather prediction.

– *AVHRR cloud fraction* test: it is based on the cloud fraction value derived from the collocated AVHRR imager in the IASI FOV. This test is positive (a cloud is present) if the cloud fraction exceeds 2%.

– *AVHRR heterogeneity* test: it is based on the analysis of the heterogeneity in collocated AVHRR images.



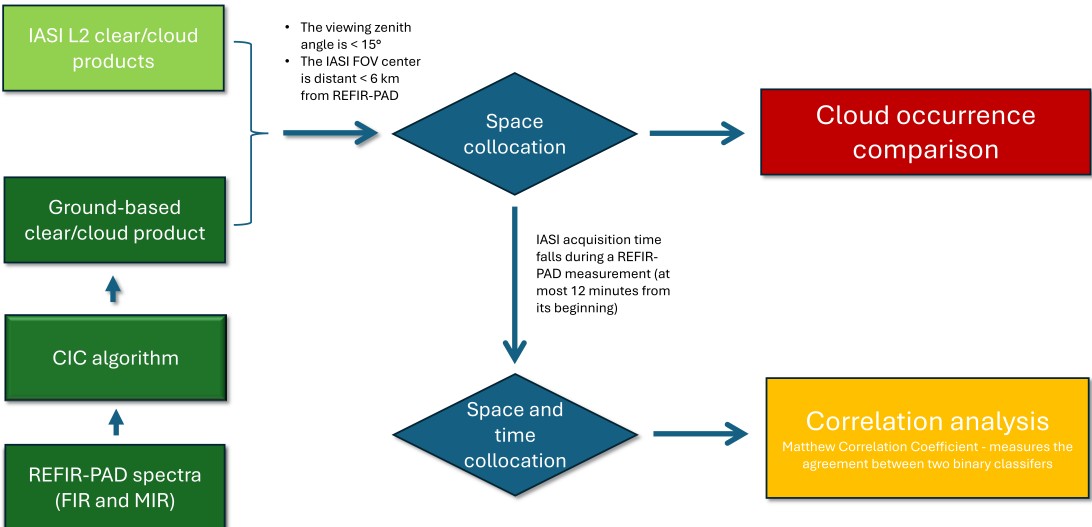

**Figure 2.** Workflow of the process of comparison between the IASI L2 cloud products with the CIC/REFIR-PAD cloud products.

- *ANN* test: it is based on the application of an artificial neural network to both the IASI radiance and the collocated
AVHRR measurements in combination.

In addition to cloud tests, a cloudiness summary is provided in the *cldnes* flag. The flag can take the following values:

- 1: The FOV is clear.

- 2: Small cloud contamination is possible.

- 3: The FOV is partially covered by clouds.

- 4: High or full cloud coverage.

The value of *cldnes* depends on the results of the *NWP*, *AVHRR cloud fraction*, and *ANN* tests, as well as on the outcomes of cloud top pressure retrieval ($CO_2$-slicing method) and cloud fraction retrieval ($\chi^2$ method). Based on the value of the *cldnes* flag, we derive three different clear/cloud products:

- The FOV is considered cloudy if *cldnes* is 2, 3, or 4 (clear conservative).

- The FOV is considered cloudy if *cldnes* is 3 or 4.

- The FOV is considered cloudy if *cldnes* is 4 (cloud conservative).



Finally, we also account for the flag named *cloud phase*. This flag takes value equal to 0 when the FOV is considered clear, 1 in the case of liquid cloud, 2 for ice cloud, and 3 for mixed-phase cloud. The result is obtained from the analysis of brightness temperature differences: $T_b(8\ \mu m)$ - $T_b(11\ \mu m)$ and $T_b(11\ \mu m)$ - $T_b(12\ \mu m)$. The clear/cloud occurrence is defined so that the
FOV is cloudy if *cloud phase* $> 0$.

## 2.3 REFIR-PAD and IASI L2 products correlation analysis

Matthews Correlation Coefficient is an index commonly used to measure the agreement between two binary classifiers (Chicco and Jurman, 2020). The MCC definition is:

$$MCC = \frac{n_{11} \cdot n_{00} - n_{10} \cdot n_{01}}{\sqrt{(n_{11} + n_{10}) \cdot (n_{01} + n_{00}) \cdot (n_{10} + n_{00}) \cdot (n_{01} + n_{11})}} \in [-1, 1], \tag{1}$$

where

- $n_{11}$ is the number of scenes identified as cloudy by both the ground and the satellite classifier;

- $n_{00}$ is the number of scenes identified as clear by both the ground and the satellite classifier;

- $n_{10}$ is the number of scenes identified as cloudy by the ground classifier and as clear by the satellite classifier;

- $n_{01}$ is the number of scenes identified as clear by the ground classifier and as cloudy by the satellite classifier.

A positive MCC indicates that the two classifiers tend to put the colocated scene into the same class while a negative MCC indicates the opposite. In particular, MCC = +1 means perfect agreement, and MCC = -1 means that the ground and satellite classifications are always different. Note that MCC is not defined if one of the factors in the denominator is zero, i.e. if one of the two classifiers sees only clear skies or only cloudy skies. In the present study, the MCC index is used only on spatially and temporally collocated IASI/REFIR-PAD observations.

## 2.4 Assessment of data consistency

The comparison between products series derived from ground-based and satellite data might be affected by dissimilarities in the time and spatial resolutions of the two sensors. Protocols and tools, such as the EUMETSAT MONALiSA ("MONitoring of Atmospheric Level 2 SAtellite products"), are defined mostly for the validation and calibration of clear sky products against ground-based observations. When cloudy sky observations are considered, the satellite and ground-based comparison is made
more difficult by inherent small scale variability of cloud properties. In this case tools with limited applications are available such as the EUMETSAT METIS-C https://metis.eumetsat.int/index.html. We are aware of possible inconsistencies and the following comparisons are provided within the limits of the discussed differences in the measurement characteristics which have been minimized by strict time and space collocations between the sensors. In this Section, we examine the impacts for the differences in the sensors' FOVs and measurement time (sampling and dwell time) on the results of the comparison between
products derived from IASI and from REFIR-PAD. Our comparison strategy relies on very conservative requirements for the





time and space collocation parameters which are usually less stringent when clear sky data are considered. The adopted strategy is made possible by the long record available for both ground and satellite measurements. The differences in the sensed air masses are minimized by accounting for satellite measurements over Dome C with observational angles smaller than $15°$. For the one-to-one comparison between IASI and REFIR-PAD products, only time collocated observations are considered as described in Figure 2.

Compensation effects reducing possible differences in the ground-based and satellite measurements might exist. In fact, it has been demonstrated that the CIC is highly sensitive to the presence of clouds Cossich et al. (2021) Magurno et al. (2020) and capable to identify very thin scattering layers and thus ensuring very high scores in cloud detection. Nevertheless, in some situations, the CIC high sensitivity might provide a slight overestimation of cloud occurrences due to positive cloud identification in presence of "non-cloud" scattering layers such as significant diamond dust events. The diamond dust episodes are, otherwise, hardly identified by satellite mostly due to their small optical depths and the similarities in their thermodynamic property with the surface. A possible overestimation of cloud identification by IASI with respect to REFIR-PAD consists in the largest size of the satellite FOV which in some situations might include clouds that are not entering the FOV of the ground-based sensor. We note, also, that the largest time required by REFIR-PAD to perform a measurement could lead to the acquisition of radiance from varying scenes (clear to cloud or viceversa) which are likely seen as cloudy by the CIC algorithm. Thus, the largest acquisition time of REFIR PAD sensor with respect to IASI is compensating for its smaller FOV.

### 2.4.1 IASI sampling time

Since the satellite overpass on the ground station occurs only within specific time frames, there might be possible biases in cloud occurrence statistics. For this reason, we evaluate if the ground-based cloud occurrence product, obtained applying the CIC on REFIR-PAD observations, would significantly change if the considered observations refer to the time-frames of the IASI overpasses only instead of the whole dataset. Note that a possible bias due to a non-uniform time coverage of the IASI observations does not affect the correlation analysis since it is computed for temporally collocated observations only.

In Figure 3 the hourly distribution of the IASI observations over Dome C and of the REFIR-PAD observations are reported. Panel (a) shows the non-uniform distribution of IASI overpass on Concordia Station that is characterized by peaks at around 15:00 UTC and 23:00 UTC. In contrast, REFIR/PAD observations are uniformly distributed along the day. However, the difference does not significantly affect the cloud occurrence product and the comparison with the CIC/REFIR-PAD data remains valid. In the same Figure, the CIC/REFIR-PAD monthly cloud occurrences from 2014 to 2020 (lower left panel) and their monthly mean (lower right panel) are shown for the full dataset and for times restricted to the IASI overpasses time intervals (14:00-17:00 UTC and 21:00-23:59 UTC). The comparison shows that minimal differences in cloud occurrence are found (with the highest values limited to less than 10%), and, more importantly, temporal trends are preserved.

### 2.4.2 Dwell time and spatial resolution

An additional source of uncertainty affecting the comparison between the IASI and REFIR-PAD products derives from the different times required to perform a single observation. In fact, in case of IASI the dwell time is about 8 seconds which is





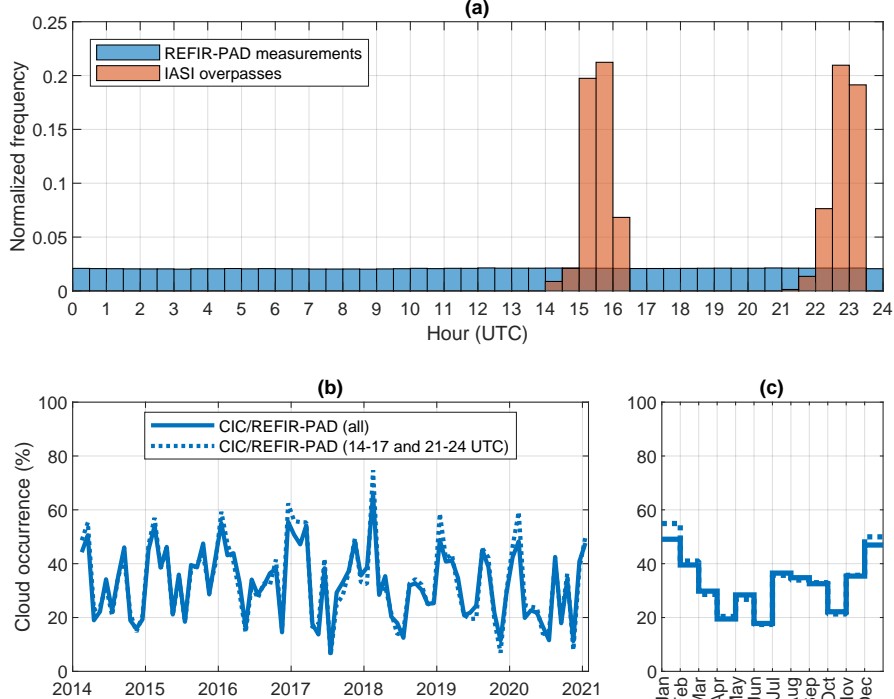

**Figure 3.** Upper panel: hourly normalized distribution of REFIR-PAD observations (blue) and IASI overpasses (red) at Concordia Station during the period 2014 to 2020. Lower panels: panel (b) shows monthly cloud occurrence from CIC/REFIR-PAD data using the full dataset (solid line) and observations restricted to IASI overpass times (dashed line). Panel (c) presents the same comparison, but for monthly mean values.

significantly smaller than the REFIR-PAD measurement time. The ground-based measurement is derived from averaging 8
210 spectra acquired in about 12 minutes. This implies that during a REFIR-PAD measurement it is statistically more likely that
the observed scene undergoes a modification with respect to what could occur during the IASI measurement. An analysis of
the persistence of the cloudy scenes, based on the CIC identification, is performed. Results show that more than 70% of cloudy
scenes last for more than 30 minutes suggesting that the scene conditions vary slowly with time. If changes in scene condition
might affect the REFIR-PAD statistic is difficult to be evaluated, but we think that it could provide a (slight) overestimation of
215 the cloud occurrence and somehow compensate for the differences in the spatial resolution.

The IASI footprint is approximately 12 km in diameter at nadir while the REFIR-PAD angular aperture of about 115 mrad
implies a FOV of about 575 m in presence of a cloud placed at 5 km above the ground level. The FOVs' difference favours
a larger probability of cloud presence for IASI than for REFIR-PAD. Since the effects of the differences in dwell time and





FOV are difficult to be quantified, we developed a strategy based on the MCC relying on temporally and spatially collocated
observations rather than cloud occurrence time series. Under the assumptions that:

- CIC/REFIR-PAD is a perfect scene classifier (meaning it is capable to detect any cloud that enters its FOV during the
  measurement time),

- a large number of collocated IASI observations with REFIR-PAD measurements is available,

- the probability of having a cloud in the IASI FOV is larger when a cloud is detected by the REFIR-PAD during its 12
  minutes observation than when it is not,

it can be shown that a negative MCC is equivalent to a satellite clear/cloud Hit Rate (relative to the true atmopheric state in
the satellite's FOV) below $\frac{1}{2}$. The problem is formalized and the statement is demonstrated in the appendix A. In this way, a
theoretical upper bound can be placed on the satellite's classification performances using an assumed, ground truth.

## 3 Results

### 3.1 Ground-based cloud occurrence frequency analysis

|  | **All seasons** (%) | **DJF** (%) | **MAM** (%) | **JJA** (%) | **SON** (%) |
|---|---|---|---|---|---|
| Clear sky | 67.5 | 54.7 | 74.3 | 70.5 | 70.4 |
| Ice clouds | 30.0 | 35.2 | 25.6 | 29.5 | 29.0 |
| Mixed-phase clouds | 2.5 | 10.0 | 0.1 | 0.0 | 0.6 |
| Unclassified | 0.0 | 0.1 | 0.0 | 0.0 | 0.0 |

**Table 1.** CIC/REFIR-PAD classification results for the whole timeframe 2014–2020 and by trimester.

From the CIC classifications of 232,999 REFIR-PAD spectra collected between 2012 and 2020, a clear/cloud product is
derived. Since the distribution of measurements is dense and uniform only starting from year 2014 we focus in the 2014-2020
timespan. The general classification results are reported in Table 1, where it is derived that an average cloud occurrence of 30%
is found with oscillations during the year. The cloudiest season is Summer (DJF) with a total of more than 40% and the least
cloudy season is Autumn (MAM).

The results of the time series analysis are reported in Figure 4. The left panels of the figure show the yearly average cloud
occurrence (upper panel), temperature (mid panel) and pressure (lower panel). Temperature and pressure are derived from in-
situ 2005-2023 data (Grigioni et al., 2022). The three variables present a major peak in the warm season (between December
and January) and a minor peak in the cold season (between July and August). The central column of the same Figure shows
the monthly time series of cloud occurrence, temperature and pressure during the timeframe 2014-2020. Execution of the Wald
test indicates that no significant linear trend is present (a p-value of 0.05 was adopted as a threshold). The panels on the right





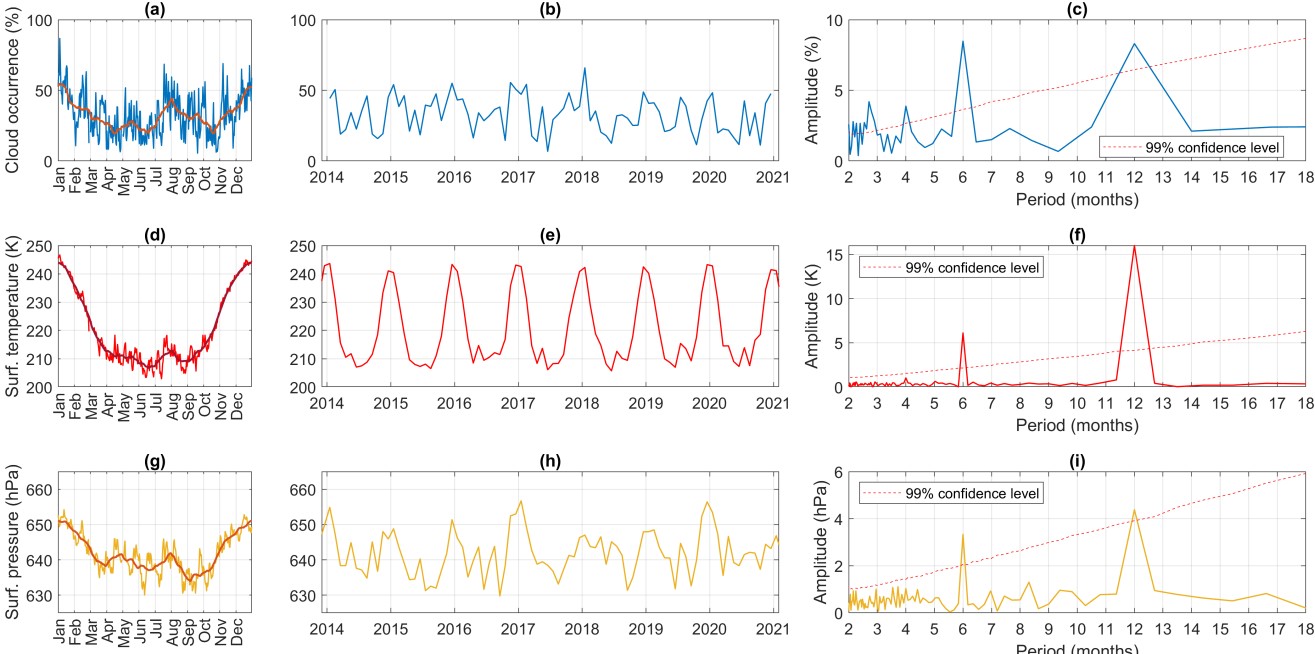

**Figure 4.** Average intra-annual trends (left column), monthly time series 2014–2020 (mid column) and relative harmonic spectra (right column) of cloud occurrence (first row), surface temperature (second row) and surface pressure (last row) at Concordia Station. Average intra-annual trends have a time step of one day and are shown together with their 31-days moving average. The time period of the spectra is cutted at 18 months since no significant peak is found beyond that period value.

column of the Figure present the amplitude of the harmonic components of the time series, computed by means of discrete Fourier transform. A 99% confidence level (dashed red line) was computed, derived from 10,000 simulations of red noise

$$x(t) = a \cdot x(t-1) + b \cdot \epsilon, \tag{2}$$

245 where $\epsilon \sim \mathcal{N}(0,1)$ and the parameters $a$ and $b$ are estimated directly from the data as explained in the appendix B. The harmonic analysis shows that cloud occurrence, temperature, and pressure share significant harmonics with periods of 12.0 months (annual cycle, peaking in December or January) and 6.0 months (semiannual cycle, peaking in December or January and in June or July). As shown in Table 2, the annual temperature cycle is significantly more pronounced than the semiannual cycle (16.0 K amplitude versus 6.1 K), while the two cycles for cloud occurrence and pressure are comparable (8.3% versus

250 8.5% for cloud occurrence, and 4.4 hPa versus 3.3 hPa for pressure). In addition, the cloud occurrence data exhibit other cycles, of minor intensities, with periods of 4.0 and 2.7 months. The existence of a 6-months surface temperature and pressure oscillation (called semiannual oscillation) was reported by Van Den Broeke (1998) after the analysis of various Antarctic stations. The amplitude and initial phase found by Van Den Broeke (1998) are consistent with our findings, and their causes are discussed in Meehl (1991). In the work of van den Broeke (2000) it is shown that a firmly established half-yearly wave in

255 the mean annual cycles of wind speed and cloudiness is found at the Antarctic Stations of Halley and Faraday. In our data the





**Cloud Occurrence**

| Period (months) | Amplitude (%) | Initial phase | Peaking time |
|---|---|---|---|
| 12.0±1.1 | 8.3±1.4 | 0.5±0.2 | 17 Dec±10 days |
| 6.0±0.6 | 8.5±1.4 | -0.1±0.2 | 19 Jul±5 days |
| 4.0±0.8 | 3.9±1.4 | 1.2±0.4 | 23 Apr±9 days |
| 2.7±0.5 | 4.2±1.4 | 2.6±0.3 | 01 Mar 2014±8 days |

**Temperature**

| Period (months) | Amplitude (K) | Initial phase | Peaking time |
|---|---|---|---|
| 12.0±0.1 | 16.0±0.4 | 0.3±0.0 | 27 Dec±1 days |
| 6.0±0.2 | 6.1±0.4 | 0.6±0.1 | 31 Jun±2 days |

**Pressure**

| Period (months) | Amplitude (hPa) | Initial phase | Peaking time |
|---|---|---|---|
| 12.0±0.9 | 4.4±0.6 | 0.0±0.1 | 16 Jan±8 days |
| 6.0±0.6 | 3.3±0.6 | 0.5±0.2 | 01 Jul±5 days |

**Table 2.** Period, amplitude, phase and peaking time of the most significant harmonics of cloud occurrence, temperature and pressure. Only 1 peaking time is indicated and subsequent peaking times can be obtained by adding a number of months equal to the corresponding period $T$. Peaking times are obtained from initial time $t_0$, phase $\phi$ and period $T$ as $t_0 - \frac{\phi}{2\pi}T$. Here $t_0$ is the 15th of January since we are dealing with monthly averages starting from January. Uncertainties are derived as in Montgomery and O'donoghue (1999).

semiannual southern oscillation is neatly observed in the cloudiness (cloud occurrence), suggesting a possible influence on the cloud radiative effect.

Figure 5 reports time series of REFIR-PAD radiances integrated over the FIR part of the spectrum which is here taken as the 100-667 cm$^{-1}$ interval. The integrated radiance obtained from all scenes (upper left panel) peaks in December/January and in July/August as expected from the cloudiness maxima occurring in the same months (Figure 4). However, while cloud occurrence peaks have similar amplitudes, the peak in downwelling radiance is much more intense in summer than in winter, likely due to the higher temperatures and, to a less extent, to an increased presence of cloudiness. It is noted that a local maximum is found in winter also for clear sky conditions only, which means that the maximum is partially independent of the increase in cloudiness. The differences between the all sky FIR integrated radiance and the one in clear sky conditions only is shown in panel (d). The difference is indicative of the cloud radiative effect at FIR. Summer and winter maxima are observed also in this case suggesting the role of clouds in enhancing the longwave radiative forcing at the ground. The panels of the mid and right columns show the FIR downwelling radiance in the presence of ice or mixed-phase clouds only with respect to the clear sky. It is shown that the highest values are found in the presence of mixed phase clouds. In case of ice clouds the most significant radiative forcing occurs in summer (when the temperatures are the highest) but the largest differences with





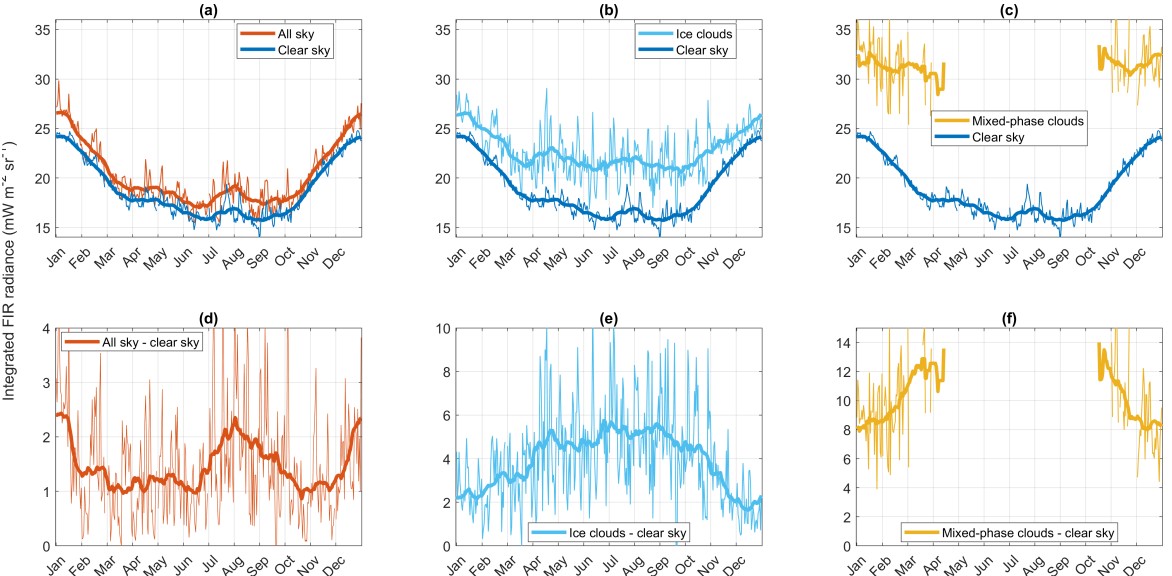

**Figure 5.** Upper panels: Downwelling integrated radiance in the FIR (100-667 cm-1) in all sky and clear conditions (left upper panel). In the other columns the radiance in clear sky is compared to that observed in the presence of ice clouds (mid upper panel), and mixed-phase clouds (right upper panel). Lower panels: radiance differences corresponding to the curves above. The 31-days moving averages are also reported for all quantities.

respect to the clear conditions are found in winter (when the atmosphere is generally cold and dry and its radiative effect is less significant).

## 3.2 Comparison with IASI products

A total of 1287 IASI observations over Concordia Station are found during the timespan from 2014 to 2020. Among these, 930 FOVs are also time collocated with REFIR-PAD acquisitions. The monthly distributions of the IASI and IASI/REFIR-PAD collocated acquisitions during the considered time interval is summarized in Figure 6.

The results obtained from the analysis of IASI L2 tests are presented in Figures 7 to 10. In Figure 7 the cloud occurrence derived by CIC applied to the REFIR-PAD data and the one from the *NWP* test on IASI are computed. Cloud occurrences are calculated based on the number of clear-sky and cloudy-sky classifications, excluding cases where IASI collocation is available but the test was not executed. In panel (a), a comparison is provided for monthly mean values over the 2014-2020 time frame. The IASI *NWP* test mostly overestimates CIC occurrences up to year 2020. In panel (b), the monthly mean cloud occurrences are averaged over the 2014–2019 period and in panel (c) the same quantity is presented only for the year 2020. The lower panels show the correlations between the two binary classifiers for the specified time periods. The correlations are determined using the MCC formula 1, applied to the binary (clear/cloud) collocated classifications from CIC/REFIR-PAD and the corresponding IASI product. The analysis reveals that the *NWP* clear/cloud classifications are generally anticorrelated with the ground-based



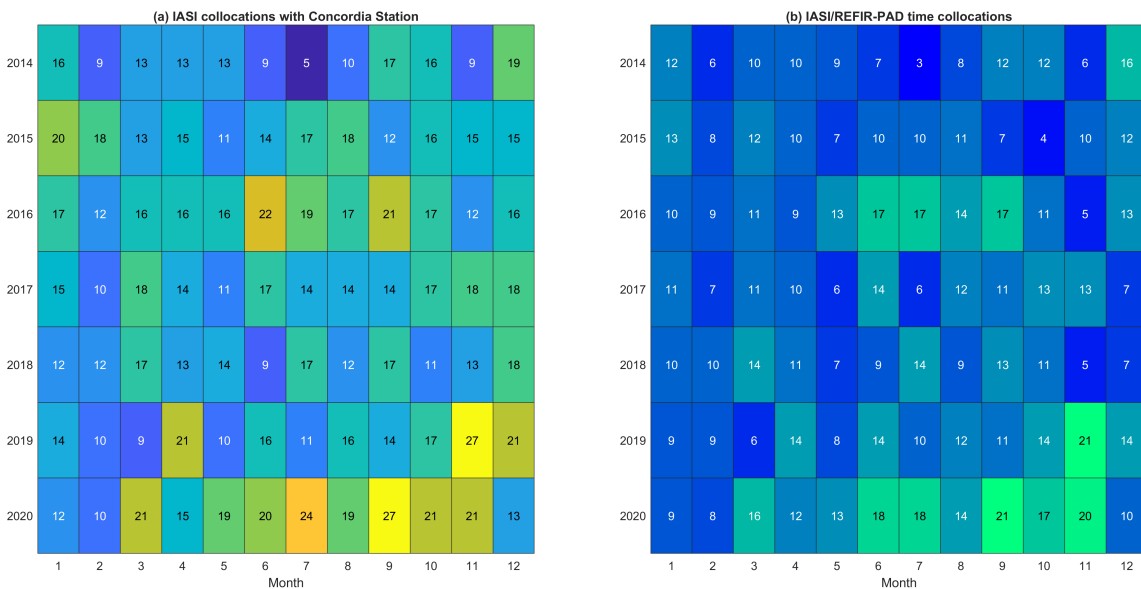

**Figure 6.** Left panel: Number of IASI observations over Concordia Station for each months from year 2014 to 2020. Right panel: Collocated IASI/REFIR-PAD monthly acquisitions in the timespan 2014-2020.

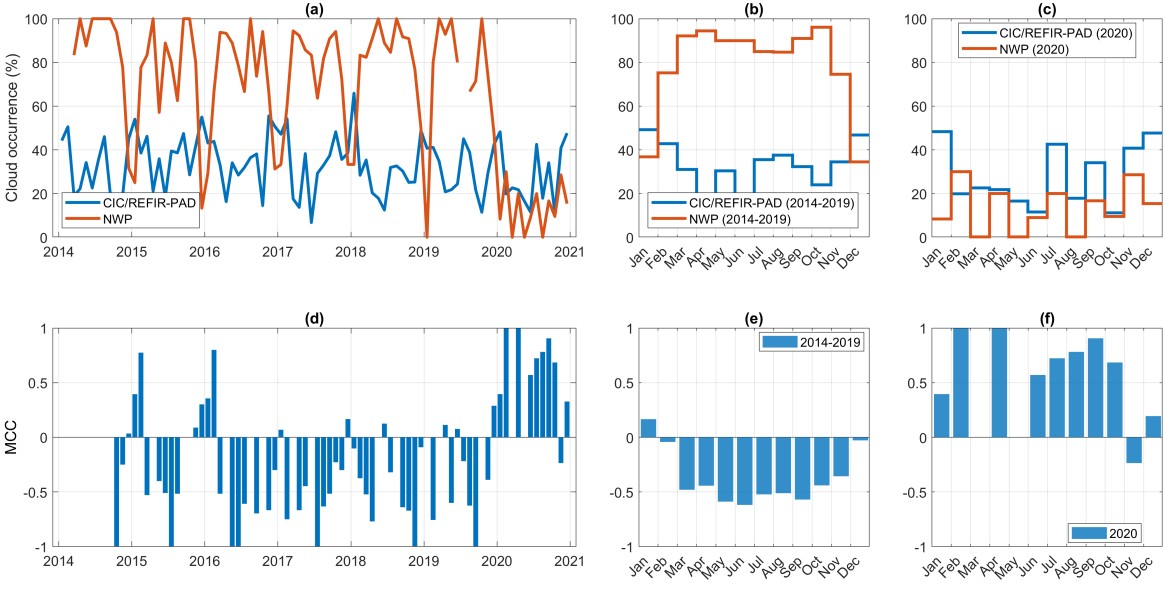

**Figure 7.** Comparison between CIC/REFIR-PAD and IASI *NWP* test. Upper panels: monthly mean cloud occurrences from 2014 to 2020 (left), averaged monthly means over the 2014-2019 timespan (middle) and monthly occurrences in year 2020 (right). The lower panels show the MCC correlations between the temporally collocated clear/cloud classifications in the same periods defined for cloud occurrence.





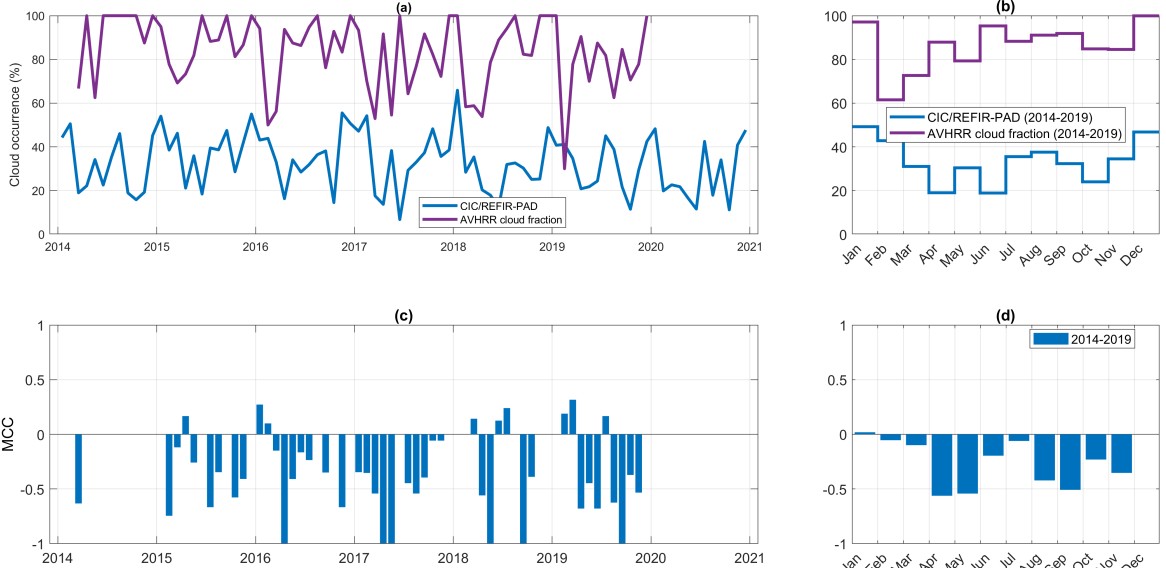

**Figure 8.** Comparison between CIC/REFIR-PAD and IASI *AVHRR cloud fraction* test. Upper panels: monthly mean cloud occurrences from 2014 to 2020 (left) and averaged monthly means over the 2014-2019 timespan (right). The lower panels show the MCC correlations between the temporally collocated clear/cloud classifications in the same periods defined for cloud occurrence.

classifications up to 2019 and abruptly shift to positive correlation in January 2020. Before 2020, the NWP cloud occurrence is in phase opposition with CIC/REFIR-PAD, with higher cloud occurrence (>80%) in the cold months. The *NWP* test provides peaks in April and October (in correspondence to CIC minima) and minima in August and December (in correspondence to CIC maxima). The correlations are weak from December to February, and negative from March to November. In 2020, the cloud occurrence derived from satellite is more consistent with the ground-based observations, and notably, the satellite classifier shows positive correlation with the CIC product derived from ground observations.

In Figure 8 a similar comparison is shown that accounts for the IASI *AVHRR cloud fraction* test. The Figure is lacking of the panels for year 2020 since the test results are not available for that year. Again, the satellite test shows an overestimation of the detected cloud occurrence over all the timespan considered and an anti-correlation with the ground classifications is present. The monthly averages of cloud occurrences derived from ground-based data have minima in July and maxima in December-January. For the same periods *AVHRR cloud fraction* test derives very high values of cloudiness.

The *AVHRR heterogeneity* test (Figure 9) shows a positive correlation with CIC/REFIR, even if the derived cloud occurrence values are generally underestimated. The IASI monthly averaged cloud occurences obtained from the *AVHRR heterogeneity* test peak in April. The test classifies the scenes as clear in nearly all observations between October and January. This result is in contrast to the ground classification obtained by CIC, which detects the highest cloud occurrence values during the antarctic summer.




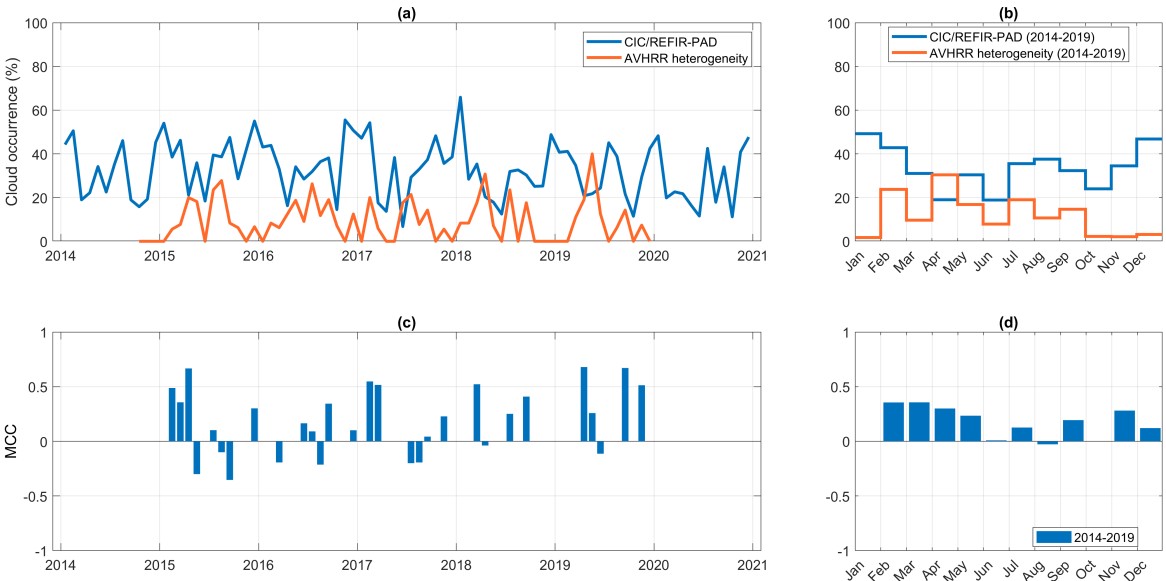

**Figure 9.** Comparison between CIC/REFIR-PAD and IASI *AVHRR heterogeneity* test. Upper panels: monthly mean cloud occurrences from 2014 to 2020 (left) and averaged monthly means over the 2014-2019 timespan (right). The lower panels show the MCC correlations between the temporally collocated clear/cloud classifications in the same periods defined for cloud occurrence.

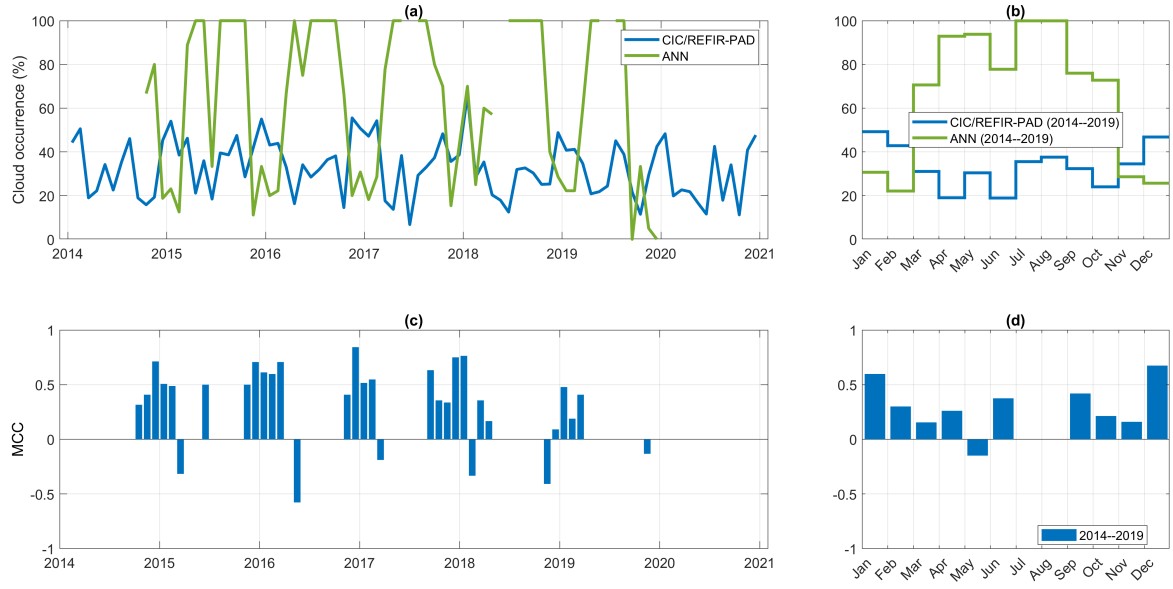

**Figure 10.** Comparison between CIC/REFIR-PAD and IASI *ANN* test. Upper panels: monthly mean cloud occurrences from 2014 to 2020 (left), averaged monthly means over the 2014-2019 timespan (middle) and monthly occurrences in year 2020 (right). The lower panels show the MCC correlations between the temporally collocated clear/cloud classifications in the same periods defined for cloud occurrence.





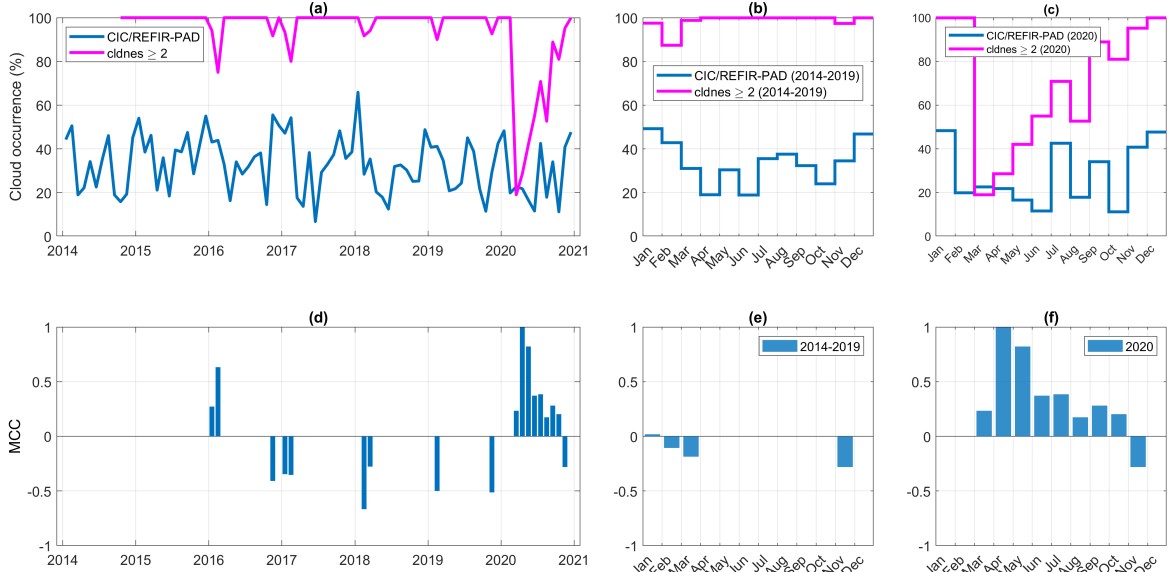

**Figure 11.** Comparison between CIC/REFIR-PAD and IASI *cldnes* product. Upper panels: monthly mean cloud occurrences from 2014 to 2020 (left), averaged monthly means over the 2014-2019 timespan (middle) and monthly occurrences in year 2020 (right); the IASI FOV is assumed as cloudy if *cldnes* ≥ 2. The lower panels show the MCC correlations between the temporally collocated clear/cloud classifications in the same periods defined for cloud occurrence.

The *ANN* test (Fig. 10) overestimates the cloud occurrence during the cold season by detecting almost always cloudy scenes in July and August. A secondary maximum is found in April-May while minima are expected in the warm season when CIC provides the highest cloud occurrences. During summer, satellite and ground-based values are comparable (even if the *ANN* test underestimates the occurrences) and a positive correlation is found on IASI and REFIR-PAD collocated observations.

Similar analysis are conducted for two additional products: *cldnes* and *cloud phase*. As described in Section 2.2.1 the *cldnes* is a composite product obtained by combining multiple cloud tests and characterized by a 4-values index. A clear conservative setting is to assume that the scene is cloudy when the index *cldnes* = 2 (small cloud contamination), 3 (partial cloud covered) or 4 (overcast). In this case very high cloud occurrences are found as shown in Figure 11. The plot shows that systematically higher cloud occurrence values are found with respect to the CIC/REFIR-PAD results. Averaged monthly minima occur in

February and November and the scene is classified always as cloudy from April to October. However, in March 2020 the cloud occurrence drops to match CIC/REFIR-PAD values, and the product is characterized by the same maxima (July, September and end of the year 2020) that are recognized also from ground identifications. Consistently, in 2020 the satellite classifications are positively correlated with the ground-based ones.

A less clear conservative classification is obtained assuming the scene as cloudy if *cldnes* = 3 or 4. The results of the

comparison are presented in Figure 12. It is shown that up to year 2019 the satellite monthly mean cloud occurrence oscillates between 90% and 20%, with minima in January and maxima in June. The classifications are usually anti-correlated with



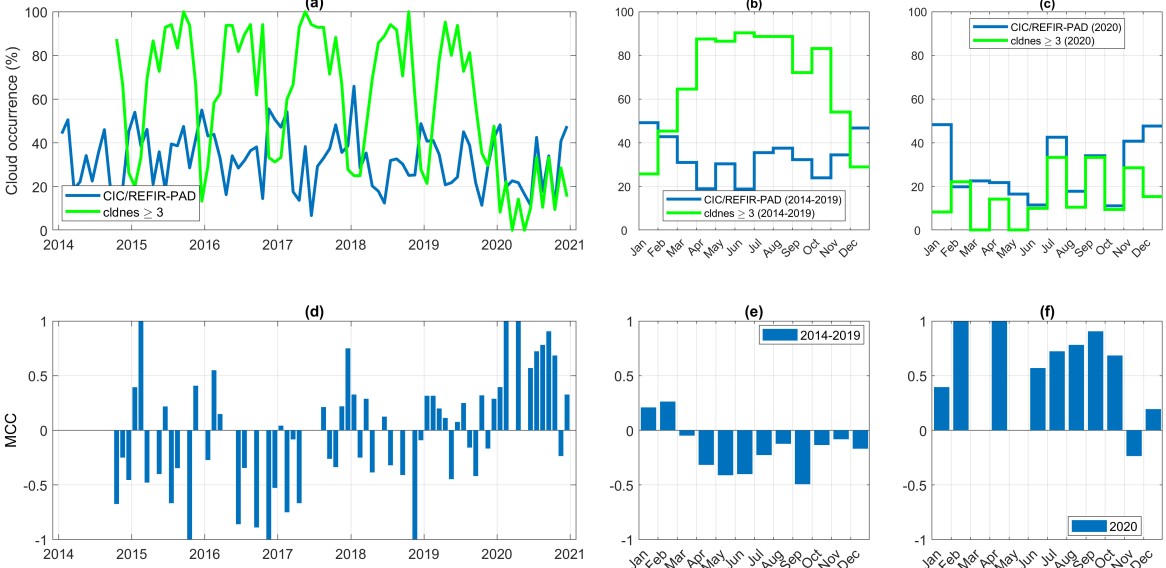

**Figure 12.** Comparison between CIC/REFIR-PAD and IASI *cldnes* product. Upper panels: monthly mean cloud occurrences from 2014 to 2020 (left), averaged monthly means over the 2014-2019 timespan (middle) and monthly occurrences in year 2020 (right); the IASI FOV is assumed as cloudy if *cldnes* ≥ 3. The lower panels show the MCC correlations between the temporally collocated clear/cloud classifications in the same periods defined for cloud occurrence.

collocated CIC/REFIR-PAD, except that in January and February. In 2020 the oscillation observed in data referring to the period 2014-2019 is not present, and the cloud occurrence values align more closely with the ground values, with classifications showing improved correlation.

When assuming the scene cloudy for *cldnes* = 4 (meaning detection of overcast conditions), the satellite cloud occurrence largely underestimates the results found from ground. The comparison is shown in Figure 13. The product is uncorrelated with collocated data analyzed by CIC and no significant differences are observed for data processed before and after 2020.

A final comparison considers the cloud occurrence derived from the flag *cloud phase*. For any values larger than 0 (meaning clear sky) it is assumed that a cloud is present in the IASI observation (*cloud phase* > 0). The results are presented in Figure
14. From March 2012 to September 2014, the satellite product indicates the presence of clouds in every scene (only a part of the timespan is shown in the plot). From October 2014 to 2019 the cloud occurrence values are more coherent with what found from ground observations; nevertheless they are mostly overestimating the CIC derived cloud occurrences. Averaged monthly mean data show the highest values during the Antarctic spring and minima in Fall. The correlations of the average monthly means are positive but small. From year 2020 the satellite product (probably refined) agrees better with the CIC cloud
identification and the correlation values largely increase. An excellent agreement is found between the two products during the July-October 2020 timeframe. Table 3 summarizes the overall correlation values between temporally collocated CIC/REFIR-PAD and IASI products. For the products available, the period before 2020 and year 2020 are considered separately to highlight





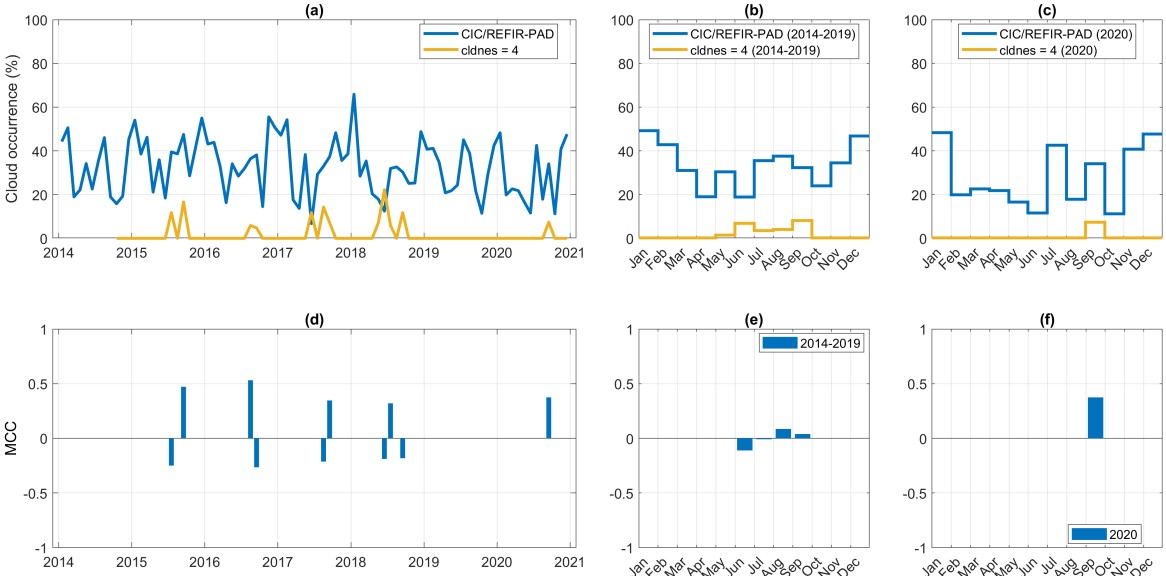

**Figure 13.** Comparison between CIC/REFIR-PAD and IASI *cldnes* product. Upper panels: monthly mean cloud occurrences from 2014 to 2020 (left), averaged monthly means over the 2014-2019 timespan (middle) and monthly occurrences in year 2020 (right); the IASI FOV is assumed as cloudy if *cldnes* = 4. The lower panels show the MCC correlations between the temporally collocated clear/cloud classifications in the same periods defined for cloud occurrence.

| IASI L2 clear/cloud classifier | MCC 2014–2019 | MCC 2020 |
|:---:|:---:|:---:|
| NWP | −0.36 (613) | +0.41 (138) |
| AVHRR cloud fraction | −0.23 (723) | - (0) |
| AVHRR heterogeneity | +0.14 (663) | - (0) |
| ANN | +0.36 (376) | - (0) |
| cldnes≥2 | −0.05 (677) | +0.32 (176) |
| cldnes≥3 | −0.20 (677) | +0.51 (176) |
| cldnes = 4 | −0.01 (677) | +0.16 (176) |
| cloud phase≥1 | +0.10 (821) | +0.51 (175) |

**Table 3.** IASI L2 clear/cloud products and correlation values (MCC) with the temporally collocated CIC/REFIR-PAD classifications. In parentheses, the number of scenes used for the computation is reported. For the product *cloud phase* only scenes from October 2014 are considered since the product is always cloudy up to that date.

the increased performances of the satellite products. For year 2020, it is shown that the best performances are obtained using the IASI products *cldnes* with values ≥ 3 and *cloud phase* with values ≥1.





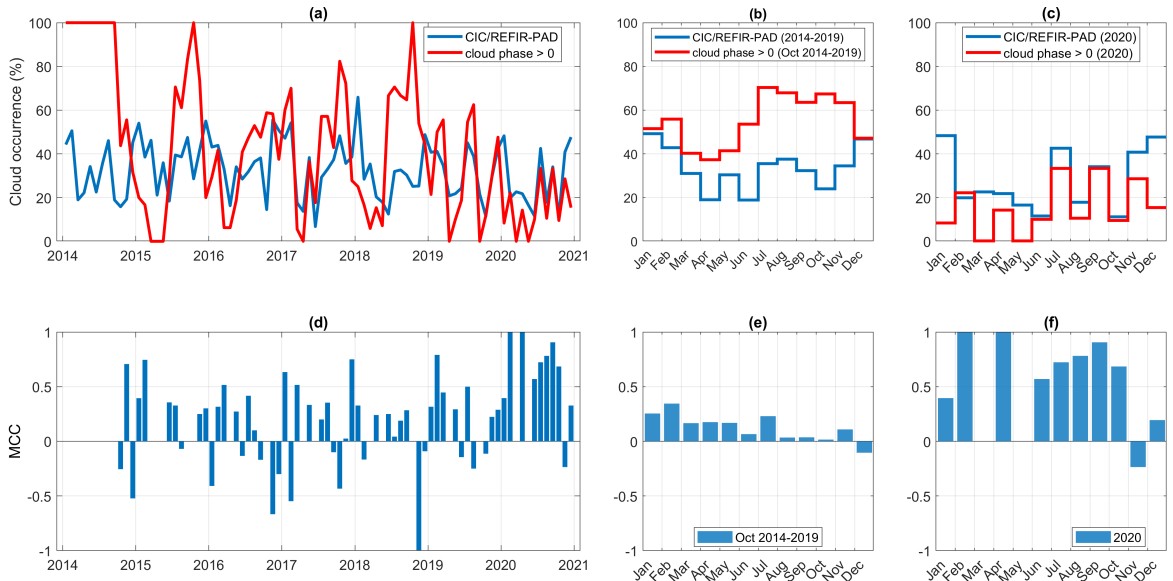

**Figure 14.** Comparison between CIC/REFIR-PAD and IASI *cloud phase* product. Upper panels: monthly mean cloud occurrences from 2014 to 2020 (left), averaged monthly means over the 2014-2019 timespan (middle) and monthly occurrences in year 2020 (right); the IASI FOV is assumed as cloudy if *cloud phase* > 0. The lower panels show the MCC correlations between the temporally collocated clear/cloud classifications in the same periods defined for cloud occurrence.

### 3.2.1 Classification of thermodynamic phase

Out of 997 IASI/REFIR-PAD temporally colocated scenes from 2012 to 2020, 179 scenes are identified as cloudy by both CIC/REFIR-PAD and the IASI flag *cloud phase*. Table 4 summarizes the phase classifications for these 179 scenes.

|  | IASI ice clouds | IASI mixed or liquid phase clouds |
|---|---|---|
| CIC/REFIR-PAD ice clouds | 167 | 0 |
| CIC/REFIR-PAD mixed-phase clouds | 8 | 4 |

**Table 4.** CIC/REFIR-PAD classification results for the whole timeframe 2014–2020 and by trimester.

IASI indicates the presence of liquid water only if CIC/REFIR-PAD does the same, while some clouds classified as mixed by CIC/REFIR-PAD are classified as purely icy by IASI. An overall correlation (MCC) was computed considering the classes "ice cloud" and "mixed-phase or liquid cloud". The resulting correlation score is 0.56, indicating that IASI recognizes the same cloud phase as CIC/REFIR-PAD in scenes where both classifiers indicate a cloud. CIC/REFIR-PAD detects mixed-phase clouds during the warm season each year from 2012 to 2020. However, IASI mixed-phase cloud classifications are missing in



the summer season of 2015/2016, 2016/2017, and 2018/2019. For both classifiers, the peaks in mixed-phase cloud occurrence
are observed between December and January.

## 4 Summary

An extensive dataset, consisting of approximately 233,000 ground-based "clear sky", "ice cloud" and "mixed-phase cloud" clas-
sifications from 2012 to 2020, is generated by applying the Cloud Identification and Classification (CIC) algorithm to spectrally
resolved radiance measurements performed by the Far InfraRed - Prototype for Applications and Development (REFIR-PAD)
sensor in the far infrared and mid infrared part of the spectrum. This dataset is used to compute cloud occurrence time series
at Concordia Station, Dome C, in the middle of the Antarctic Plateau. The cloud occurrence exhibits clear seasonal patterns,
with prominent maxima between December and January, and secondary peaks between July and August. Harmonic analysis
reveals that the cloud occurrence is composed of an annual cycle (12 months, maxima in January) and a semi-annual cycle
(6 months, maxima in January and July). These significant cycles are also observed in the collocated surface temperature and
pressure data, displaying similar initial phases. The six-month cycle is known, in the literature, as the semi-annual Southern
Oscillation. By analysing the downwelling radiances in all sky conditions and in clear conditions only, it is retrieved that the
far infrared cloud radiative effect during the peaks of the semi-annual oscillation is about twice as high as during its minima.
The ground-based dataset is also used to test satellite cloud products derived from infrared passive observations. In this regard,
the Infrared Atmospheric Sounding Interferometer (IASI) cloud products are considered with the goal of evaluating the effi-
ciency of satellite infrared sounders in identifying cloudy scenes at polar latitudes. Several IASI L2 cloud tests are compared
with ground-based products. The IASI flag *cldnes*, which provides a cloudiness summary ("clear", "possible cloud contami-
nation", "partial cloud coverage", or "full cloud coverage"), is transformed into three different clear/cloud products adopting
increasingly clear-conservative cloud definitions. Similarly, the IASI flag *cloud phase*, that reports the result of a cloud phase
retrieval ("clear", "liquid cloud", "ice cloud", or "mixed-phase cloud"), is considered as a clear/cloud product indicating the
presence of a cloud when a phase is successfully retrieved. A correlation analysis is performed for temporally collocated clas-
sifications using the Matthews Correlation Coefficient (MCC), which evaluates the agreement between two binary classifiers.
The correlations are computed in three ways: (1) monthly, to assess possible seasonal patterns; (2) as a time series through-
out the entire study period; and (3) globally, across all data. This correlation-based approach allows us to verify whether the
identification of a cloud (or a clear sky) from the ground is statistically associated with a higher (or lower) probability of cloud
detection by the satellite, compared to the baseline probability. In addition, this methodology helps address issues caused by
the differing spatial and temporal resolutions of the ground-based and the satellite sensors. Specifically, the MCC provides a
way to establish a theoretical upper bound on the satellite's clear/cloud Hit Rate relative to the atmospheric state within the
satellite's FOV. Assuming the ground-based product serves as a perfect binary classifier for the REFIR-PAD FOV, we show
mathematically that a negative MCC implies a satellite Hit Rate below $\frac{1}{2}$, whereas a positive MCC indicates a Hit Rate larger
than $\frac{1}{2}$. The results indicate that, prior to 2020, specific clear/cloud products (*NWP*, *AVHRR cloud fraction*, flag *cldnes*) are anti-
correlated with the ground-based data, as they tend to classify the same scene in the opposite way; the strongest anticorrelations



(MCC$< -0.40$) are found in the cold months. In the period 2014-2019 the test *AVHRR heterogeneity* and the flag *cloud phase* present low values in the overall correlations (respectively +0.14 and +0.10) with CIC/REFIR-PAD, although many months are characterized by $\approx 0$ correlation. *ANN* is globally well correlated (+0.36), especially in January and December where values overcome +0.50, although it classifies all scenes as cloudy in July and August. Cloud occurrence trends derived from

these products were analyzed. Expectedly given the lack of strong positive correlation throughout the whole year, 2014-2019 monthly cloud occurrences derived from these products are generally inconsistent with the ground-based cloud occurrence. For instance, *NWP*, *ANN*, and the *cldnes* flag show cloud occurrence minima when the ground-based data indicate maxima (during the warm season), and vice versa, with maxima corresponding to minima during the cold season.

In 2020, the available products include *NWP*, flag *cldnes*, and flag *cloud phase*. Interestingly, the correlation values for *NWP*

and flag *cldnes* shift abruptly from negative to positive. Notably, *NWP* passes from -0.36 to +0.41, while the flag *cloud phase* increases the positive correlation from +0.10 to +0.51. The IASI cloud occurrences more closely mirror ground-based observations in 2020 than in the period 2012–2019.

Finally, we found that the flag *cloud phase* indicates the presence of mixed-phase or liquid clouds especially in December and January, in agreement with CIC/REFIR-PAD. A one-to-one comparison, made on the 179 scenes where both the IASI flag

*cloud phase* and CIC/REFIR-PAD indicate the presence of a cloud, reveals that IASI detects the presence of liquid water only if CIC/REFIR-PAD does the same from the ground. MCC applied to these 179 "ice cloud" or "mixed-phase or liquid cloud" classifications yields a value of +0.56, meaning that the two classifiers are in good agreement for the cloud thermodynamic phase.

*Code and data availability.* The CIC source code version used in the present paper is available on request to the corresponding authors.

Meteorological data are available at http://www.climantartide.it (last access: 12 June 2025). REFIR-PAD radiances can be acquired from https://refir.fi.ino.it/products.html. IASI Level 2 Data can be downloaded from the EO:EUM:DAT:METOP:IASSND02 collection using the EUMETSAT Data Access Client (EUMDAC) application.

## Appendix A: Theoretical upper bound on satellite Hit Rate

Let us consider the comparison between the cloud classification performance of two instruments: the satellite-based IASI and

the ground-based CIC/REFIR-PAD, which observes in a zenith-looking configuration. IASI and CIC/REFIR-PAD differ both in spatial and temporal resolution. In particular, the field of view (FOV) of IASI is significantly larger than that of CIC/REFIR-PAD:

$$FOV_{CIC} \subset FOV_{IASI}. \tag{A1}$$

As a result, a cloud could be present in $FOV_{IASI}$ but not in $FOV_{CIC}$. Similarly, the instruments differ in temporal resolution.

REFIR-PAD collects measurements over a 12-minute time interval, denoted by

$$\Delta T = [T_1, T_2], \tag{A2}$$





while IASI provides an observation over a much shorter time interval

$$\delta t = [t_1, t_2],\ \delta t \subset \Delta T. \tag{A3}$$

We denote the durations of these intervals as

$\quad |\delta t| = t_2 - t_1 \ll |\Delta T| = T_2 - T_1 = 12\,\mathrm{min}.$ (A4)

We define the temporal overlap between the two instruments as

$$\delta T_{overlap} = \Delta T \cap \delta t, \tag{A5}$$

with duration $|\delta T_{overlap}| = |\delta t|$. Due to this difference in temporal resolution, a cloud may be present in $FOV_{CIC}$ during a part of $\Delta T$ but not during $\delta T_{overlap}$, possibly due to formation/dissipation or advection.

Despite these factors, we show that a theoretical upper bound can be placed on IASI's classification performance using MCC. The result holds under simple assumptions and explicitly accounts for differences in spatial and temporal resolution. First we set a cloud definition and consider the following:

- $HR_0$: hit rate for correct clear-sky classification by IASI,

- $HR_1$: hit rate for correct cloudy-sky classification by IASI,

- $\mathbb{P}(1_{\Delta T}^{FOV_{CIC}})$: unconditional probability that a cloud is present in $FOV_{CIC}$ during $\Delta T$

- $\mathbb{P}(1_{\delta T_{overlap}}^{FOV_{IASI}})$: unconditional probability that a cloud is present in $FOV_{IASI}$ during $\delta T_{overlap}$.

- $\mathbb{P}(1_{\delta T_{overlap}}^{FOV_{IASI}}|0_{\Delta T}^{FOV_{CIC}})$: probability that a cloud is present in $FOV_{IASI}$ during $\delta T_{overlap}$, given that no cloud is present in $FOV_{CIC}$ during $\Delta T$

- $\mathbb{P}(1_{\delta T_{overlap}}^{FOV_{IASI}}|1_{\Delta T}^{FOV_{CIC}})$: probability that a cloud is present in $FOV_{IASI}$ during $\delta T_{overlap}$, given that a cloud is present in $FOV_{CIC}$ during $\Delta T$


We now assume:

1. The REFIR-PAD classifier (e.g., CIC/REFIR-PAD) reliably distinguishes between clear and cloudy conditions in its FOV,

2. A large number $N \gg 1$ of collocated IASI/REFIR-PAD observations is available,

3. $\mathbb{P}(1_{\delta T_{overlap}}^{FOV_{IASI}}|1_{\Delta T}^{FOV_{CIC}}) > \mathbb{P}(1_{\delta T_{overlap}}^{FOV_{IASI}}|0_{\Delta T}^{FOV_{CIC}}).$

Under these assumptions, it can be proved that a negative MCC between the two classifiers implies

$$HR = \frac{HR_0 + HR_1}{2} < \frac{1}{2}, \tag{A6}$$

the Hit Rate being against the actual atmospheric state in $FOV_{IASI}$.





## A1 Demonstration

For hypotheses 1 and 2 the number of scenes classified as clear by CIC/REFIR-PAD can be written as $N \cdot \mathbb{P}(0_{\Delta T}^{FOV_{CIC}})$. The number of scenes classified as clear by both CIC/REFIR-PAD and IASI depends on the conditioned probabilities $\mathbb{P}(0_{\delta T_{overlap}}^{FOV_{IASI}}|0_{\Delta T}^{FOV_{CIC}})$ and $\mathbb{P}(1_{\delta T_{overlap}}^{FOV_{IASI}}|0_{\Delta T}^{FOV_{CIC}})$, as well as on the hit rates $HR_{cle}$ and $HR_{cld}$:

$$n_{00} = N\mathbb{P}(0_{\Delta T}^{FOV_{CIC}})\left[\mathbb{P}(0_{\delta T_{overlap}}^{FOV_{IASI}}|0_{\Delta T}^{FOV_{CIC}})HR_0 + \mathbb{P}(1_{\delta T_{overlap}}^{FOV_{IASI}}|0_{\Delta T}^{FOV_{CIC}})(1 - HR_1)\right]. \tag{A7}$$

Similarly we have

$$n_{01} = N\mathbb{P}(0_{\Delta T}^{FOV_{CIC}})\left[\mathbb{P}(1_{\delta T_{overlap}}^{FOV_{IASI}}|0_{\Delta T}^{FOV_{CIC}})HR_1 + \mathbb{P}(0_{\delta T_{overlap}}^{FOV_{IASI}}|0_{\Delta T}^{FOV_{CIC}})(1 - HR_0)\right], \tag{A8}$$

$$n_{11} = N\mathbb{P}(1_{\Delta T}^{FOV_{CIC}})\left[\mathbb{P}(1_{\delta T_{overlap}}^{FOV_{IASI}}|1_{\Delta T}^{FOV_{CIC}})HR_1 + \mathbb{P}(0_{\delta T_{overlap}}^{FOV_{IASI}}|1_{\Delta T}^{FOV_{CIC}})(1 - HR_0)\right], \tag{A9}$$

$$n_{10} = N\mathbb{P}(1_{\Delta T}^{FOV_{CIC}})\left[\mathbb{P}(0_{\delta T_{overlap}}^{FOV_{IASI}}|1_{\Delta T}^{FOV_{CIC}})HR_0 + \mathbb{P}(1_{\delta T_{overlap}}^{FOV_{IASI}}|1_{\Delta T}^{FOV_{CIC}})(1 - HR_1)\right]. \tag{A10}$$

By definition $MCC < 0$ is equivalent to

$$n_{11} \cdot n_{00} < n_{01} \cdot n_{10}. \tag{A11}$$

Substitution and direct computation provides

$$(HR_0 + HR_1 - 1)\left[\mathbb{P}(1_{\delta T_{overlap}}^{FOV_{IASI}}|1_{\Delta T}^{FOV_{CIC}}) - \mathbb{P}(1_{\delta T_{overlap}}^{FOV_{IASI}}|0_{\Delta T}^{FOV_{CIC}})\right] < 0. \tag{A12}$$

For hypothesis 3 it is $\mathbb{P}(1_{\delta T_{overlap}}^{FOV_{IASI}}|1_{\Delta T}^{FOV_{CIC}}) - \mathbb{P}(1_{\delta T_{overlap}}^{FOV_{IASI}}|0_{\Delta T}^{FOV_{CIC}}) > 0$, so we conclude

$$\frac{HR_0 + HR_1}{2} < \frac{1}{2}. \tag{A13}$$

## Appendix B: Red noise parameters estimation

Modeling the data $y(t)$ as

$$y(t) = a \cdot y(t-1) + b \cdot \epsilon, \tag{B1}$$

where $\epsilon \sim \mathcal{N}(0,1)$, by multiplying both members by $y(t-1)$ and averaging on $t$ we get

$$\overline{y(t)y(t-1)} = a \cdot \overline{y(t-1)^2} + b \cdot \overline{\epsilon \cdot y(t-1)}. \tag{B2}$$

The second term on the right vanishes if the total number of time steps is $\gg 1$ since $\epsilon$ is normally distributed around 0, so for 455 $a$ we obtain

$$a = \frac{\overline{y(t)y(t-1)}}{\overline{y(t-1)^2}}. \tag{B3}$$



Squaring both members of B1 and averaging:

$$\overline{y(t)^2} = a^2 \cdot \overline{y(t-1)^2} + b^2 \cdot \overline{\epsilon^2} + 2 \cdot a \cdot b \cdot \overline{\epsilon \cdot y(t-1)}. \tag{B4}$$

Knowing that $\overline{\epsilon^2} = 1$ and that the last term vanishes, we obtain

$$b = \sqrt{\overline{y(t)^2} - a^2 \cdot \overline{y(t-1)^2}} = \sqrt{\overline{y(t)^2} \cdot (1-a^2)}. \tag{B5}$$

*Author contributions.* Conceptualization: TM; methodology and generation of the ground products: FD, EF, TM, MM; software and analyses: FD (ground and satellite) and EF (ground); download and preparation of satellite data: FD; writing: FD and TM; REFIR-PAD and lidar data: GB, MDG, GDN, LP; revision: all authors.

*Competing interests.* The authors declare that they have no conflict of interest.

*Acknowledgements.* The authors wish to thank Paolo Ruggieri for the useful discussions on Southern Hemisphere Semi-annual Oscillation. Meteorological dataset (temperature, pressure) and information are performed by the Italian Antarctic Meteo-Climatological Observatory (IAMCO) https://www.climantartide.it in the framework of the PNRA/IPEV 'Routine Meteorological Observation at Station Concordia' project.



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
