# Peer review of "Ground-based detection of Antarctic clouds: analysis of cycles and comparison with IASI products"

_EGUsphere, 2025_

## Author Comment (AC1)

Reply to Reviewer 1.

Reviewer comments are highlighted in yellow

1. Even though each topic (climatology and satellite validation) are, in principle interesting, they are not developed with the necessary depth to warrant publication in AMT. First, the findings on cloud annual cycle and the six-month cycle are well known, as the authors properly documented. The analysis of the far-infrared radiation is interesting, but it does not meaningfully add to what is known about cloud and climate variability of Antarctica. Moreover, these types of analyses fall outside the scope of AMT: "The main subject areas comprise the development, intercomparison, and validation of measurement instruments and techniques of data processing and information retrieval for gases, aerosols, and cloud."

We respectfully disagree with the reviewer for several reasons.

Our primary contribution lies in the intercomparison of cloud mask products derived from IASI against the REFIR-PAD ground-based spectroradiometer, which aligns directly with AMT's focus on the development, intercomparison, and validation of measurement techniques. The satellite intercomparison includes over 1,200 collocated observations with rigorous statistical analysis using MCC methodology. The manuscript presents the first application of a new metric for cloud identification and classification to ground-based observations. The metric represents an improvement over the previous version; its detailed description was omitted here for brevity, as it has already been introduced in a referenced SPIE conference proceeding (Donat et al., 2024). In the revised manuscript, we will provide a fuller description of the metric and explicitly demonstrate the improvements relative to the previous approach. We believe this contribution fits well within the main subject areas of AMT, as it illustrates the development and intercomparison of a data-processing technique for cloud analysis.

About cloud and climate variability in Antarctica, we showed a set of new results. We regret that they have been missed by the reviewer. Our climatology analysis spans 8 years with 233,000 measurements - one of the most extensive ground-based cloud datasets from Antarctica. The amount of data alone warrants the required depth, exceeding that of several existing studies on the same topic.

- A summary of the original result is reported:
  - The Antarctic plateau is underexplored and the paper provides detailed statistics of cloudiness based on an extensive period of observations, which represent a unique set of retrieval
  - The cloud cycles are evaluated, for the first time, for the Antarctic plateau, and their amplitude is estimated as the same amount.
  - Till now, a cycle in cloudiness has been observed (Van Den Broeke, 2000) only in coastal stations which can hardly be assimilated to Concordia in terms of meteorological conditions.

- Moreover, in the cited study the cloudiness is quantified in octas, thus considering all the visible sky, and presumably, without any automatic classification algorithm.
- Two cycles of minor intensities are discovered: 4.0 and 2.7 months.
  They have never been observed before.
- The analysis of the FIR radiation reveals that cloud forcing at the surface follows a semiannual cycle, which represents a novel finding. Furthermore, the forcing during summer and winter is of comparable magnitude—a result not previously reported. This is particularly surprising given that the dominant cloud types differ between the two seasons (ice clouds in winter versus mixed-phase clouds in summer), as highlighted by the reported statistics in Cossich et al. 2021.
- Our analysis demonstrates for the first time that FIR cloud radiative effect during semiannual peaks is approximately twice that during minima (Figure 5). This quantification is critical for understanding polar cloud radiative forcing and has direct implications for climate model validation in polar regions, as shown by Di Natale et al. (2017) who demonstrated the unique capabilities of FIR spectroscopy for cirrus cloud property retrieval that cannot be achieved with traditional methods.

Please also note that the manuscript shows that cloud occurrence cycles are captured consistently by a ground-based spectroradiometer using a dedicated identification/classification algorithm, thus confirming the robustness of the method. It also highlights how the REFIR-PAD—derived climatology compares with satellite products, revealing systematic differences that are directly relevant for IASI users and algorithm developers. The climatological context is essential for demonstrating instrument performance across seasonal cycles, which is standard practice in AMT intercomparison papers.

The aspects of novelty of the manuscript will be better highlighted in the revised version and discussed in the conclusions.

2. The validation of IASI products is better aligned with AMT's subject areas. However, as a validation paper, the structure and analysis better match those for a conference proceeding than an article ready for peer review analysis. Generally speaking, the analysis is primarily the comparison of mean values and (Matthews) correlation coefficient, without an in-depth analysis of the causes for the discrepancy or how the analysis could inform algorithm developers or IASI users about best practices for using the satellite products in atmospheric research. Along this line, the repetition of findings in a summary section, and the exclusion of a discussion and conclusion sections, make this reviewer think that the authors rushed the submission of their manuscript.

We would like to clarify that the manuscript does not present a validation study of IASI products. Rather, it focuses on an intercomparison among different methods and observational configurations. The term validation appears only once in the manuscript, and solely in reference to the MONALiSA protocol, cited in the Assessment of data consistency section, to highlight that existing community protocols are currently available only for clear-sky comparisons.

We do indeed provide a first analysis of discrepancy causes, namely: (1) FOV differences and their compensation effects (Section 2.4), (2) temporal resolution impacts (Section 2.4.1-2.4.2), (3) algorithm-specific performance patterns, and (4) product evolution over time (dramatic improvement in 2020). An in-depth analysis, however, would go beyond the scopes of the paper, which is the intercomparison between ground-based and satellite products in the context of Antarctic Plateau.

However, the analysis is not restricted to the mere calculation of mean values and Matthews correlation coefficients. Indeed, we developed a theorem to formally obtain, from MCC values, benchmarks on the satellite performances against the satellite field of view. Specifically, we stated in Section 2.4.2 and demonstrated in the Appendix A that negative (positive) MCC values indicate satellite Hit Rate below (above) 50%. This information is valuable both for IASI users, who can better understand the behavior of the products in a polar environment, and for algorithm developers, who can use these results to refine retrieval strategies. To make this contribution more evident, in the revised manuscript, we will extend the discussion to highlight methodological and practical implications for users and developers.

Specifically, we will indicate which products we recommend on the Antarctic Plateau for the periods 2014-2019 (when not all products have good scores based on our analyses) and in 2020. Also, we will highlight that seasonal performance varies significantly - warm season scores are consistently higher across all products, suggesting algorithm optimization for polar winter conditions is needed.

3. "Regarding the analysis, I argue that it is not possible to obtain statistically meaningful results by analyzing correlations or mean values at a monthly scale. From Fig. 6, especially for the spatiotemporal collocation, the number of samples per month are generally less than 14 samples. With just a few samples, it is very likely that the statistical values of MCC in Figs. 7 are negligible, and thus, no inferences should be made in terms of long-term relationships (times series) between REFIR-PAD and IASI."

We acknowledge this concern and wish to clarify our methodology. Our MCC analysis is performed aggregately over the 2014–2019 and 2020 timeframes (Table 3), based on hundreds of collocated classifications, showing dramatical improvement for those products that are available in both periods ('NWP' passes from -0.36 to +0.41, 'cldnes>=2' from -0.05 to +0.32, 'cldnes>=3' from -0.20 to +0.51, 'cldnes=4' from -0.01 to +0.16, as well as the flag 'cloud phase', that passes from +0.10 to +0.51). Monthly presentation serves mostly to visualize this temporal pattern rather than for individual statistical inference. However, we agree with the reviewer that the limited monthly

sampling is a challenge. In the revision we will show the 2014-2020 MCC time series aggregating data at a seasonal and yearly scale to demonstrate that sample size does not affect the results (like the sudden improvement in 2020). We already showed data that way in the 2024 IASI conference with a presentation named 'IASI cloud detection on the Antarctic Plateau and comparison with ground-based interferometric measurements', obtaining the same long-term behaviour. This will ensure that our conclusions are properly contextualized in terms of statistical significance.

4. "As the authors stated, the different spatial resolution of IASI products and the ground-based observation is an important impediment. The fundamental question here is whether REFIR-PAD retrievals are well suited for validating satellite products with a footprint of several km. And this is not a minor concern, as the FOV of REFIR-PAD is not even 5% the scale of the pixel resolution of IASI. Sometimes, the high sampling rate of ground-based instruments like lidars (ceilometer), can help translate temporal statistics to spatial ones, under the assumption of constant wind. This is more challenging to do with REFIR-PAD, but it could be attempted. Regrettably, I am not convinced that this validation is methodologically sound. At least, the alternative hypothesis that the discrepancy between ground-based and satellite data is attributed to the dissimilar sampling characteristics cannot be falsified."

As stated before, this paper is about comparison between ground and satellite products. The reviewer's logic would invalidate the entire field of satellite validation, which routinely uses point measurements (radiosondes, AERONET, ground stations) to validate satellite retrievals with kilometer-scale footprints. More importantly, the effectiveness of comparison - not validation - between satellite and ground-based products depends on methodology rather than geometric similarity. The 'dissimilar sampling characteristics' hypothesis for negative correlation values is falsified by our MCC theorem, explicitly accounting (appendix A) for scale differences. As already stated, we provide mathematical proof that negative (positive) MCC values indicate satellite hit rates against the satellite FOV below (above) 50% regardless of sampling differences. This implies that negative correlation values require an 'algorithm performance' explanation. However, this does not mean that dissimilar sampling has no effect at all - if that was the case, we would have been able to place exact values on the satellite Hit Rate rather than simple (however meaningful) bounds.

5. Why do some statistics improve after 2019? Again, this is a critical question that needs to be addressed.

We thank the reviewer for raising this critical point. The exact cause cannot be firmly established with the public documentation available.

Since the ground cloud identification procedure is always the same, the systematic improvement across multiple IASI products in 2020 clearly indicates IASI operational processing system changes. However, no documentation is available covering the 2020

IASI products. We will explicitly acknowledge this as a limitation and recommend that future studies address this issue in collaboration with the data provider (EUMETSAT).

6. If the goal is to evaluate sounder retrievals, then what is the purpose of including products derived from a broader band instrument such as AVHRR.

The reviewer should be aware that the IASI Level 1C (L1C) data are provided along with the AVHRR cloud mask from the same Metop platform. Since the two instruments fly aboard the same satellite, the AVHRR cloud mask, which has a significantly finer spatial resolution than IASI, constitutes a crucial aid for the quality control and processing of the IASI data.

I am not providing a list of edits and suggestions at this time because I anticipate significant changes in a revised manuscript. That being said, the manuscript needs to be carefully proofread.

While we note that a more detailed list of edits and suggestions would have been helpful for addressing specific issues, we will thoroughly revise the manuscript for clarity and grammar. We will make substantial revisions to improve readability and presentation throughout the text.

**References:**

Cossich Marcial De Farias W.; Maestri T.; Magurno D.; Martinazzo M.; Di Natale G.; Palchetti L.; Bianchini G.; Del Guasta M., *Ice and mixed-phase cloud statistics on the Antarctic Plateau*, «ATMOSPHERIC CHEMISTRY AND PHYSICS», 2021, 21, pp. 13811 - 13833. https://doi.org/10.5194/acp-21-13811-2021

Federico Donat, Elisa Fabbri, Tiziano Maestri, Michele Martinazzo, Fabrizio Masin, Giorgia Proietti Pelliccia, Lorenzo Cassini, Guido Masiello, Giuliano Liuzzi, Carmine Serio, "The cloud identification and classification (CIC) algorithm for high spectral resolution observations in the far- and mid-infrared part of the spectrum," Proc. SPIE 13193, Remote Sensing of Clouds and the Atmosphere XXIX, 1319302 (20 November 2024); https://doi.org/10.1117/12.3031709